# Influence of Graphene Oxide on Mechanical Properties and Durability of Cement Mortar

**DOI:** 10.3390/ma17061445

**Published:** 2024-03-21

**Authors:** Lounis Djenaoucine, Álvaro Picazo, Miguel Ángel de la Rubia, Amparo Moragues, Jaime C. Gálvez

**Affiliations:** 1Departamento de Ingeniería Civil: Construcción, E.T.S.I. Caminos, Canales y Puertos, Universidad Politécnica de Madrid, C/Profesor Aranguren 3, 28040 Madrid, Spain; lounis.djenaoucine@alumnos.upm.es (L.D.); miguelangel.rubia@upm.es (M.Á.d.l.R.); amparo.moragues@upm.es (A.M.); 2Departamento de Tecnología de la Edificación, E.T.S. de Edificación, Universidad Politécnica de Madrid, Avda. Juan de Herrera, 6, 28040 Madrid, Spain; a.picazo@upm.es

**Keywords:** graphene oxide, cement, mechanical properties, durability

## Abstract

The effect of graphene oxide (GO) on the mechanical strengths and durability of cement composites was researched by preparing GO-modified cement mortars. Thermogravimetric analysis (TGA) and nuclear magnetic resonance (^29^Si MAS-NMR) were performed on the cement paste to evaluate the influence of GO on the hydration process and chain structure of calcium-silicate-hydrate (C–S–H) gels. TGA revealed that the high GO dosage increased the content of C–S–H by 5.46% compared with the control at 28 days. Similarly, ^29^Si-NMR improved the hydration degree and main chain length (MCL) in GO-modified samples at 28 days. The GO led to increases of 2.54% and 7.01% in the hydration degree and MCL, respectively, compared with the control at 28 days. These findings underscore the multifaceted role of GO in improving the mechanical properties and durability of cement composites. Mechanical strength tests, such as compressive and flexural tests, were conducted on cement mortars. The optimal dosage of GO increased the compressive strength by 9.02% after 28 days. Furthermore, the flexural strength of cement mortars with the combination of GO and superplasticizer (SP) after 28 days increased by 21.86%, compared with reference mortar. The impact of GO proved to be more pronounced and beneficial in the durability tests, suggesting that GO can enhance the microstructure through hydration products to create a dense and interconnected microstructure.

## 1. Introduction

Nowadays, cementitious materials (CM) such as concrete and cement mortar are still the most widely used material in civil engineering and construction, being used for the construction of buildings, bridges, dams, and tunnels [1], among others. The two main characteristics that define the quality of CM, such as mortar, are strength and durability. Strength is the ability of mortar to withstand loads (e.g., compression and tension) and is influenced by a variety of factors, such as materials quality used, mixing method and curing process. Durability is the key factor in determining the service life of the CM, which can be defined as the mortar’s ability to withstand chemical attacks (e.g., CO_2_ and Cl^−^) and climatic conditions [2]. However, the main drawback that threatens CM is its inherent brittleness, which is due to its poor resistance to cracking, low tensile strength and high porosity, which can affect its mechanical properties and long-term durability [3,4]. In a bid to overcome the aforementioned drawbacks of CM, several studies advocate for the use of nanotechnology, which is considered one of the most reliable solutions due to the progress made and offers a tremendous opportunity to further improve the performance of CM as well as adjust its drawbacks by incorporating nanomaterials [5,6].

Nanomaterials are considered one of the most promising solutions due to their ability to reach the nanoscale level and affect the hydration products that are the main components determining the strength and durability of CM. The nanomaterials that have attracted the researchers’ interest in the civil engineering field are numerous, include nano-silica (SiO_2_) [7], nano-alumina (Al_2_O_3_) [8], carbon nanotubes (CNTs) [9] and, more recently, graphene oxide (GO) [10,11,12,13,14,15,16,17,18,19].

The graphene derivative, GO, exhibits extraordinary mechanical characteristics, including an intrinsic strength of around 130 GPa and a Young’s modulus of approximately 1 TPa [20]. However, it has a high surface area and a unique structure [21], with various oxygen-containing functional groups situated at its basal plane and edge, such as hydroxyl (–OH), epoxide ether (C–O–C), carbonyl (C=O) and carboxyl (–COOH) [22,23]. The presence of these functional groups makes GO hydrophilic and therefore easily dispersible in water. [24], For this reason, GO is considered one of the preferred nanomaterials to be incorporated into CM to form a homogeneous mixture that leads to better strength and durability [18,19,25,26]. Several papers have referred to the effects of GO-modified cement-based materials; most of these investigations are focused on mechanical strength and durability. A study conducted by Ganesh et al. [27] found that 0.03% GO is the optimal amount to enhance the mechanical properties of cement mortar. The study also found that the oxygen functional groups on the GO sheets act as nucleation sites for the formation of calcium silicate hydrate (C–S–H) and calcium hydroxide (Ca(OH)_2_) crystals. The more organized structure of the C–S–H and Ca(OH)_2_ crystals in the presence of GO results in improved strength properties. Pan et al. [17] reported that the addition of 0.05% GO sheet into ordinary Portland cement paste increased the compressive and flexural strength by 15–33% and 41–58%, respectively. The reasons for the increase are probably due to the effect of GO on the hydration products, where chemical reactions among them produce strong interfacial forces, which could also be due to a strong interaction between GO sheets and cracks due to their 2D geometry. Gong et al. [16] found that the addition of 0.03% GO sheets to hardened cement paste resulted in a 46% increase in compressive strength and the tensile strength by almost 50%, owing to the enhancement of the hydration degree and the reduction in the porosity of the hardened cement paste. Furthermore, nuclear magnetic resonance (^29^Si MAS-NMR) studies on tricalcium silicate (C_3_S) conducted by Yang et al. [15] showed that adding GO to CM increases the hydration degree which obviously leads to an increase in compressive strength. Wang et al. [14] showed that 0.03% of GO caused a significant improvement in the mechanical properties of cement mortar after 28 days, compressive, flexural and tensile strengths increased by 21.37%, 39.62% and 53.77% respectively.

As is also the case with durability, many studies have shown that the introduction of GO into CM results in varying degrees of improvement. Mohammad et al. [28] mentioned in their study that incorporating graphene oxide into a cement mortar makes it very resistant to chloride penetration due to the structure of the graphene oxide, which is intertwined and resembles a sponge structure that prevents the penetration of destructive particles such as chlorides ions. This can improve the durability of CM and increase the structures lifespan. Bagheri et al. [29] also investigated the impact of incorporating GO on the transport properties and durability of cementitious composites. Their findings revealed an enhancement in transport properties, indicating improved durability behavior, upon the introduction of GO into the mortar. This enhancement was attributed to a significant reduction in initial and secondary water absorption rate and sorptivity. The study further suggests that the amount of GO content significantly affects transport properties. Higher GO content (0.1% and above) led to a more effective reduction in porosity and voids, consequently improving water resistance. Additionally, the authors observed an increase in electrical resistivity due to the incorporation of GO, which is generally considered beneficial for concrete durability. However, they also caution that excessive GO content can negatively impact resistivity. Table 1 shows further studies.

In researching the purpose of GO-modified cement mortar strengths and durability, first, a deep look at the nanoscale level was taken to study the effects of GO on cement paste, such as hydration products and (C–S–H) gels chain structure, since these are the mains contributors to the CM performance improvement, using TGA and ^29^Si MAS-NMR characterization techniques. Subsequently, the influence of varying amounts of GO on the mechanical properties and durability of cement mortar was systematically researched. This study encompassed analyses of compressive and flexural strengths, electrical resistivity, gas permeability, and chloride diffusion. Cement mortar specimens were prepared with two different water–cement ratios: 0.50 and 0.35.

## 2. Materials and Methods

### 2.1. Materials

To carry out the present research, cement mortars were prepared, in accordance with the EN 196-1 standard [33], mixing cement, sand, SP (only in the mix with ratio *w*/*c* = 0.35) and GO to evaluate the effect of the addition of this latter component on the properties of pastes and mortars. For this purpose, Portland cement type CEM I 52.5R with high initial strength was used, whose composition is shown in Table 2. The GO solution used was supplied by Graphenea (San Sebastián, Spain). The commercial polycarboxylate-based superplasticizer (SP) was also used to ensure a good dispersion of GO in mortars (only in mortars with a *w*/*c* ratio of 0.35). Lastly, CEN-NOMRSAND standardized sand was used for all mortars, according to the EN 196-1 standard [33], arranged in bags with 1350 ± 5 g content.

### 2.2. Preparation of GO–Water Suspension

The chosen dosages of Graphene Oxide (GO) are meticulously calibrated to strike an equilibrium between the efficacy of dispersion and cost-effectiveness. This approach ensures the judicious use of GO, thereby preventing agglomeration and promoting practicality. The dispersion of GO within the cement matrix (CM) is of paramount importance due to the propensity of GO to agglomerate rapidly within the matrix, particularly when the water-to-cement (*w*/*c*) ratio is low. This is attributed to GO’s extensive surface area and oxygen functional groups. Several researchers have devoted their efforts to this issue, leading to the consensus that the optimal method for incorporating GO into cement involves dissolving it in the mixing water initially [13]. (In the course of the investigation, GO was employed in a liquefied state suspended within an aqueous medium). The process begins with weighing the required amount of water in accordance with each dosage. Subsequently, the volume of GO solution corresponding to the weight of the dosage is pipetted and introduced into the weighed water. This GO–water suspension is then subjected to stirring for a duration of five minutes using a magnetic stirrer operating at a rotational speed of 800 revolutions per minute. In instances where samples are prepared with Superplasticizer (SP), the SP is added into the GO–water suspension and stirred for an additional minute at an unchanged rotational speed. Figure 1a provides a visual representation of the employed GO dosages; the higher the concentration of GO, the darker it becomes.

### 2.3. Preparation of Mortar and Paste Samples

As previously mentioned, in this study two *w*/*c* ratios were chosen: 0.35 and 0.50 (with and without SP, respectively). The cement-to-sand ratio (*c*/*s*) was kept at 1/3, and while the precise dosages of graphene oxide (GO) employed cannot be divulged due to confidentiality agreements with the supplier, three distinct GO dosages were utilized, denoted as G1, G2 and G3 [34]. Finally, the lowest concentration, G1, is an order of magnitude lower than G2 and two orders of magnitude lower than G3. Additionally, a control sample, labeled as G0, was prepared without the inclusion of GO to serve as a benchmark for comparison. Table 3 and Table 4 list the mixed proportions of the GO-modified cement mortar and paste that was prepared. The kneading procedure was performed according to European standard EN 196-1:2018. The prepared GO–water suspension was placed in the mixing bowl along with the required cement powder, the mixing process was run at low speed for 30 s, then immediately switched to high speed as sand casting started for another 30 s, followed by a 90-s pause, after which the mixing process started again at high speed for 60 s.

To study the effect of GO on the cement hydration process, GO-modified cement paste was prepared (*w*/*c* was kept at 0.50 for all doses as indicated in Table 4). The prepared GO–water suspension and the cement powder were placed in the mixing container, followed by kneading at low speed for 90 s. After 30 s of standing, the mixture was remixed for 90 s at low speed.

After the mixing process was completed, each fresh mixture was placed in a specific mold. The prepared mortar was placed in two types of molds. The first molds type was used to make prismatic specimens of dimensions 160 × 40 × 40 mm^3^ (for mechanical testing). The fresh mix was placed in two layers and each layer was compacted with 60 strokes using an automatic compactor. The second type of molds used for mortars were cylindrical ø75 × 150 mm^3^ (for durability testing), in which the fresh mixture was poured in two layers and vibrated on a vibrating table to ensure good compaction. In addition, the cement pastes were made in the form of prismatic bars of dimensions 60 × 10 × 10 mm^3^ and were compacted as in the case of the prismatic mortar specimens. Lastly, all molds were covered with plastic sheet and placed in a curing chamber with controlled ambient conditions (20 ± 2 °C/RH > 95%). All samples were demolded after 24 h and kept in the curing chamber at the same conditions until the test day. So, Figure 1b shows the three-specimen type and Figure 2 shows the production process.

### 2.4. Characterization Techniques

#### 2.4.1. Thermal Gravimetric Analysis (TGA)

Four cement paste samples were analyzed by thermogravimetric analysis (TG/DTG). Before the analysis, the hydration process of cement paste should be halted at the age necessary to analyze the samples. To stop the hydration process, the cement paste was subjected to vacuum for 30 min, then submerged in isopropanol for 24 h, followed by another 24 h in an oven at 60 °C. Lastly, the samples were crushed into a powder and stored until analysis day. ATD/TG profiles were recorded according to the ASTM E 1131 standard [35] using Setaram Labsys Evo TGA_DTA equipment. A total of 50–100 mg of each type of paste was heated at 10 °C/min up to 1100 °C under a reactive nitrogen gas dynamic atmosphere. The paste samples were used as representatives of the corresponding mortars to optimize the evaluation results.

#### 2.4.2. ^29^Si Magic Angle Spinning Nuclear Magnetic Resonance

The characterization of our cement paste samples (CG0 and CG05) was conducted using ^29^Si Magic Angle Spinning Nuclear Magnetic Resonance (^29^Si MAS-NMR). The samples, crushed and aged for 7 and 28 days, were analyzed using a Bruker Spectrometer model AV400 MHz WB (Bruker, Billerica, MA, USA). This spectrometer operates at a resonant frequency of 79.5 MHz. During the analysis, we maintained a spinning speed of 10 kHz and set the pulse width to 4 µs. The relaxation time used for all experiments was 5 s, assuming differential error with respect to the usual acquisition parameters. The chemical shifts of ^29^Si were calibrated with reference to Tetramethylsilane (TMS), which was set at 0 parts per million (ppm). This is a standard procedure in NMR spectroscopy as TMS is commonly used as an internal standard due to its high symmetry and low reactivity. The data were processed using the commercial NMR software MestRenova (Mnova, https://mestrelab.com/software/mnova-software/), and the curve generated was constrained by the Gauss-Lorentz function.

#### 2.4.3. Strength Measurements

The flexural and compressive strength tests of the mortars were conducted on prismatic samples with dimensions of 160 × 40 × 40 mm^3^ at various intervals: 2, 7, 28 and 90 days. These tests adhered to the procedures delineated in the EN 196-1: 2018 standard [33]. The flexural test was executed using a three-point bonding machine (Ibertest C1B1400, Madrid, Spain), as depicted in Figure 3a. Subsequently, a hydraulic machine (Ibertest model HIB 150) was utilized for the compressive testing, as illustrated in Figure 3b. For each type of mortar, three samples were tested to determine the average flexural strength. In order to comprehensively evaluate the compressive strength, the remaining halves of each aforementioned flexural specimen were then subjected to rigorous examination, resulting in an average value derived from six discrete specimens.

#### 2.4.4. Electrical Resistivity

The evolution in the electrical resistivity of mortar specimens was assessed at different curing ages (2, 7, 14 and 28 days) following the Spanish norm UNE 83988-1:2008 [36]. A Giatec RCON™ device was employed to measure the electrical resistance across a range of frequencies from 1 to 30 kHz. The electrical resistivity was then computed using Equation (1). The mean value of three replicates was reported for each type of mortar. The influence of GO on microstructural development over time was examined by monitoring the changes in electrical resistivity in the mortar specimens.

Figure 4 shows an electrical resistivity test.
(1)$$\rho_{e}=\frac{S}{L}R_{e}$$
where *ρ_e_* is the electrical resistivity (Ωm); *S* is the sample surface area over which the electric charge flows (m^2^); *L* is the specimen height (m); and *R_e_* is the electrical resistance (Ω).

#### 2.4.5. Oxygen Permeability Test

Oxygen permeability was determined on cylindrical mortar sample pieces ø75 × 50 mm^3^, according to the UNE 83981:2008 standard [37] and RILEM-CEMBureau method [38]. After 28 days of curing in the curing chamber, samples were placed in a closed container with a controlled environment (65–75% relative humidity and 20 ± 2 °C temperature) for the purpose of drying the water droplets within the pores and achieving a steady weight. The method’s principle is to apply a specified pressure of oxygen gas to quantify the gas flow time through the sample. Equation (2) was used to calculate the oxygen permeability coefficient *K* for a given pressure *P*.
(2)$$K\;(m^{2})=\frac{2\;\cdot Q\cdot \;P_{a}\;\cdot \;L\;\cdot \;\;\eta }{A\;\cdot \left(P^{2}-P_{a}^{2}\right)}$$
where *Q* is the volumetric flow rate, *P_a_* is the atmospheric pressure, *L* is the specimen thickness, *η* is the dynamic oxygen viscosity, *A* is the section of the specimen and *P* is the applied pressure.

#### 2.4.6. Chloride Diffusion Profiles

Chloride diffusion tests were performed according to Nordest NT Build 443 [39] to assess total chloride content. For the chloride penetration tests, cylindrical samples of ø75 × 150 mm^3^ were used. After 28 days of curing in a humid chamber (98% relative humidity and 20 ± 2 °C temperature), the lower and lateral surfaces of each specimen were protected with an epoxy resin, leaving the upper circular face free to ensure that Cl^−^ diffusion occurred only through the upper face. The samples were then exposed for 35 days to water with sodium chloride (165 ± 1 g NaCl/1 L). After that period, different slices of each sample were extracted, obtained in the form of powder every mm of depth (8 slices of 1 mm each, up to 8 mm of depth were obtained) from the top. A suitable grinding wheel adapted to a drilling machine (Bosch Professional CGS 28 LCE, Robert Bosch España S.L.U., Madrid, Spain) was used. Next, the extracted powder was dried at a temperature of 105 ± 5 °C for 24 h and, lastly, it was placed in a desiccator to prevent rehydration of the powder until the time of the test. The powder of these layers was then chemically analyzed for chloride concentration (total chloride) using a Metrohm model 916 Ti-Touch titrator (Metrohm, Herisau, Switzerland).

## 3. Results and Discussion

### 3.1. Effect of GO on the Cement Paste Hydration Process

The hydration process of four kinds of cement pastes (CG0, CG1, CG02 and CG3) at 7 and 28 days was studied, as well as the hydration product amount of C–S–H and CH, to address the relationship between the hydration process and mechanical strength. A thermogravimetric analysis (TGA), as well as the derivative thermogravimetric (DTG), were used to identify and quantify the phases resulting from the cement hydration, such as C–S–H gel and CH phases. According to Figure 5, the DTG curves include three significant peaks. The first peak appears in the 105–430 °C range due the loss of bound water as a result of the decomposition of calcium silicate hydrate C–S–H. The second peak corresponds to the dihydroxylation of calcium hydroxide CH that appears at 430–530 °C. The third peak, located approximately at 550–1100 °C, corresponds to the decarbonation of calcium carbonate CaCO_3_ [40,41].

Figure 6 provides an overview of the cement paste weight losses at 7 and 28 days. The quantitative study of the first and second stages revealed that GO had no impact on the quantity of C–S–H and CH formed during cement hydration on the 7 days of curing or slightly delayed cement hydration, since the C–S–H gel and portlandite it contains are a bit lower than reference.

At 28 days of age, the C–S–H amount increases with the increasing of GO dosage. The CG3 sample achieved the highest result, increasing by 5.46% in C–S–H compared with the reference sample CG0. In addition, GO does not have a significant effect on the CH amount produced even at advanced ages (28 days). Therefore, at 28 days, the inhibitory effect caused by GO during the early curing stages dissipates, hence the formation of C–S–H gel further increases.

The results can be explained by the following. At the onset of the hydration process, GO acts as a water absorber due to its large surface area and the active oxygen groups present in its layer. This water absorption delays the hydration process in the early days, However, the delay effect is minimal due to the low graphene oxide content used, GO reduces the amount of water available for reaction with the cement components. This results in a decrease in the amount of calcium silicate hydrate (C–S–H) and calcium hydroxide (CH) formed in the early days.

However, over time, at 28 days, the cement paste samples containing GO showed a higher C–S–H content than the reference cement paste sample. This can be explained by the fact that the hydration process can occur over the GO layer, as it contains water and can attract cement components. This leads to the formation of hydration products on top of the GO layer, as GO is known to act as a nucleation site [18]. This is consistent with the study by Li et al. [12] who found that the water absorbed by GO in the early days is released over time, resulting in a recovery of the C–S–H content in advanced ages.

Previous studies have reported findings consistent with those presented in this work [14,42]. GO has a slight effect on stimulating the formation of more calcium silicate hydrate (C–S–H) but has no effect on the formation of calcium hydroxide (CH).

### 3.2. ^29^Si MAS-NMR Characterization in Pastes

The hydration degree and (C–S–H) gel structure of cement pastes CG0 and CG3 were determined using ^29^Si MAS-NMR spectroscopy after 7 and 28 days to gain a deeper insight into the formation phase (C–S–H) gel, which is the main hydration product contributing to the cohesion and strength of cementitious materials [41,43]. Furthermore, this allows for the observation of GO-induced changes in the C–S–H gel structure.

According to the findings of prior studies, *Q^n^* represents the species of silicate structures (*n* = 0–4), where *Q* denotes the silicate tetrahedron and *n* denotes the number of oxygen atoms interacting with nearby tetrahedrons. The chemical shift is a measure of the local environment of the silicon atom, and it can be used to identify the specific type of calcium silicate that the silicon atom is bonded to. The shift around −68 to −74 ppm is associated to *Q*^0^, which represents isolated tetrahedra, such as alite and belite (unhydrated). The shift located at approximately −76 to −80 ppm is linked to *Q*^1^, and represents the beginning or end of silicate tetrahedra. The shift located at approximately −81.5 to −85.5 ppm is linked to *Q*^2^ which represents the center site of the silicate chain. *Q*^2^ may be further split into the bridging and pairing sites *Q*^2*b*^ and *Q*^2*p*^, respectively [44]. In this study, results comparable to previous studies were obtained [10,11].

As shown in Figure 7, the two samples, CG0 and CG3, have similar absorption peaks *Q*^0^, *Q*^1^, *Q*^2*b*^ and *Q*^2*p*^ corresponding to the identical elements of the C–S–H gel constituents. However, the changing intensities highlight the variation in the chain structure. From the data in Table 5, the mean chain length (MCL) and the hydration degree (α) of C–S–H were estimated using Equation (3) and Equation (4), respectively [10].
(3)$$MCL=\frac{2\times \left(Q^{1}+Q^{2}\right)}{Q^{1}}$$
(4)$$\alpha \;\left(\%\right)=\frac{\left(Q^{1}+Q^{2}\right)}{\left(Q^{0}+Q^{1}+Q^{2}\right)}\times 100$$
where *Q^n^* represents the integral intensities of the signals.

At 7 days, the hydration degree together with the MCL of CG0 was higher than CG3. The reason may be due to the hydrophilic nature of GO, hence at the starting point of mixing it absorbs a certain amount of water and cement components, which disrupts the hydration process. Li et al. [12] also suggested that GO inhibits the hydration process at an early stage due to its absorption of water, which leads to a decrease in the cement hydration degree.

At 28 days, the hydration degree and main chain length (MCL) of CG3 improved compared with CG0. The hydration degree of CG3 rose by 2.54% compared with CG0. As well, CG3 showed a greater MCL length of 7.01%, compared with CG0. The enhancement in the hydration degree and the main chain length can be attributed to the addition of GO, which can accelerate the hydration process of cement at advanced ages, due to its ability to act as a nucleation site for the formation and development of hydrated products.

Moreover, the percentage of unhydrated tetrahedral silica (Q^0^), in sample CG3 was 3.24% greater than in sample CG0 after 7 days. However, after 28 days, sample CG3 experienced the greatest decrease in Q^0^ (by 6.51%) compared to CG0. This suggests that the hydration process in sample CG3 was more advanced than in sample CG0 at 28 days.

In more detail, first, the high surface area of GO served as a deposition platform for the water and cement components that are adsorbed in the first days. As a result, the hydration degree increased. Second, the lengthening of the main chain MCL after the addition of GO could be due to the generation of more tetrahedron *Q*^2*b*^, as shown in Table 4, on the GO surface area, since it is known that tetrahedron *Q*^2*b*^ is bridge-shaped, and therefore can act as point connection between short chains, generating a larger chain length. It can also be seen from the results in Table 5 that the diminution in the unhydrated Q^0^(%) and the augmentation of the MCL can be attributed to the inherent properties of Q^0^(%). Upon the liberation of free water absorbed by the GO, a hydration process accelerates, transforming the Q^0^(%) into a silicate polyhedron. This transformation facilitates the adhesion to pre-existing chains, a process made possible by the bridging capabilities of the GO. Consequently, this leads to an increase in the MCL. Similar results were found by Zhao et al. [11], who reported that the ability of GO sheet to act as a nucleation site causes an increase in the main chain length of C–S–H gels, resulting in an increased polymerization degree of C–S–H gels. It may be concluded that the nucleation site effect of GO accelerated the hydration process and increased the main chain length. These outcomes are consistent with what has been stated above in the TGA results.

### 3.3. Mechanical Resistance in Mortars

The effect of GO addition on the compressive and flexural strength of cement mortars, at different ages and with two different water–cement ratios, is shown in Figure 8.

At 2 days, the addition of GO did not improve the compressive strength of either mortar (with or without SP). Furthermore, comparable results were observed in terms of flexural strength between GO-modified cement mortar samples and the reference samples. As already mentioned in the TGA results, at early age the C–S–H gel amount was low in the cement paste samples modified by GO compared with the reference sample. It is well known that hydration products especially (C–S–H) gels, play a fundamental role in determining the mechanical strength of the cement composites.

At 7 days, the compressive strength outcomes for both cement mortars (with and without SP) were not obvious yet. There were some improvements, but these were still slight, particularly in samples containing a small dosage of GO owing to the tiny quantity of water absorbed, since the bigger the dose, the more water absorbed, causing an inhibition of the hydration process. However, in contrast to compressive strength, both mortars improved in terms of flexural strength when compared with the reference sample. The samples manufactured just with GO (without SP), MG2, had the highest results, increasing by 5.63% compared with the control samples. The second mortar type (with SP), MGS3, exhibited the most significant enhancement, increasing by 13.98% relative to the reference sample. This could be ascribed to the optimal dispersion of GO in the cement matrix.

At 28 days, mechanical strength results, whether compressive or flexural, for all GO-modified cement mortar samples were greater than that of the reference sample. MG3 and MGS1 had the biggest increase in compressive strength in comparison with the reference samples, increasing by 9.33% and 8.45%, respectively. It is worth noting that sample MGS1 showed greater compressive strength than sample MGS3. This is likely because the excessive amount of SP may retard or prevent the growth of hydration products.

In terms of flexural strength, the MG3 and MGS3 loading achieved the highest flexural strength in both types of mortars (with and without superplasticizer, SP). In the type without SP, sample MG3 showed a 4.23% and 19.72% increase in flexural strength at 7 and 28 days, respectively, compared to the reference mortar MG0. Sample MGS3, which contained SP, showed a 13.98% and 30.85% increase in flexural strength at 7 and 28 days, respectively, compared with the reference mortar MGS0. These results suggest that the addition of GO can improve the flexural strength of mortar and that the presence of SP can further enhance this improvement. According to a study [45], the curved edges of GO-SP are favorable for better mechanical interlocking with the cement matrix, which consequently enhances the mechanical properties of the cement mortar.

Therefore, the GO effect acting as a nucleation site, emerges as a compelling explanation for the enhancement of mechanical strength in cement composites, particularly as they age. In the initial stages of the cement–water reaction, the incorporated GO absorbs the water and cement components, thereby decelerating the hydration process in the early days. Although hydration is already underway, it primarily occurs around the GO. However, GO provides space as a platform for hydration products formed especially from water and cement components that have been absorbed. Most research that employed GO as a reinforcing agent provided an explanation for improving the mechanical strength of cement composites through the effect of GO as a nucleation site [13,46].

### 3.4. Electrical Resistivity in Mortars

As shown in Figure 9, results on the electrical resistivity of the cement mortar samples were obtained (2, 7, 14 and 28 days). The electrical resistivity gradually increased with the increase in curing age for all mortars. The reference sample’s electrical resistivity (MG0) was determined to be 14.66, 28.75, 35.24 and 38.05 (Ω.m), while for MGS0 it was found to be 37.06, 62.49, 72.97 and 83.22 (Ω.m) at curing ages of 2, 7, 14 and 28 days, respectively. Until the 7th day, the electrical resistivity for all samples manufactured with GO for both types, with and without SP, was lower than that of the reference samples MG0 and MGS0. Therefore, as stated above in the TGA and ^29^Si MAS-NMR results, this may be because samples with the highest GO content absorb more water during the first days, resulting in a reduction in the quantity of C–S–H gels formed. From 14 days, the opposite outcomes were observed. All samples containing GO, including both types (with and without SP), showed higher electrical resistivity than the reference sample. MG3 had the highest electrical resistivity tested compared with MG0; the increase was 8.83% for 14 days of curing and 21.37% for 28 days of curing. For the second type of mortar, prepared with SP, the highest value of electrical resistivity was recorded for the MGS3 sample, and the increases were 2.04% and 11.91% for 14 and 28 days of curing, respectively.

The electrical resistivity results were consistent with those of the previously mentioned TGA and ^29^Si MAS-NMR. The higher electrical resistivity is explained by the production of more C–S–H gels at advanced ages (from 14 days), due to the nucleation effect of GO in the GO-modified cement samples, whereby hydration products such as (C–S–H) gels improve the cement matrix by plug pores, thus impeding the ionic conduction pathway as mentioned in previous studies [19,47]. The improvement in electrical resistivity is attributed to the acceleration of the cement hydration process induced by GO, as it functions as a nucleation site.

### 3.5. Gas Permeability

The gas permeability coefficients of all mortar samples are shown in Figure 10. As can be seen, the gas permeability coefficient was significantly reduced for all cement mortars containing GO. The variable was determined by testing three cylindrical pieces of mortar of each type. Compared with the control mortar, the gas permeability coefficients of MG1, MG2 and MG3 decreased by 39.00%, 70.35 and 92.86%, respectively. As for the second type made with SP, MGS1, MGS2 and MGS3 were reduced by 15.30%, 57.89% and 71.71%, respectively. The structure and volume of pores are one of the most significant aspects in microstructure that impacts the service life of cementitious materials such as concrete, and consequently, gas permeability is regarded as one of the approaches to extract information about the structure and volume of pores. The gas permeability results obtained were in good agreement with the results of the electrical resistivity test (Figure 9). As can be seen in Figure 10, the gas permeability coefficients of mortar samples modified with GO were notably low, indicating that the majority of the pores were clogged (i.e., their volume reduced, or their structure altered). It is well known that clogged pores impede the ionic conduction path, and thus increase electrical resistivity. Safarkhani et al. [48] investigated the influence of GO on concrete permeability. Their findings revealed a significant reduction in permeability due to the pore size refinement and pore-filling effect of GO. Additionally, GO physically fills interstitial space, further decreasing permeability. Previous results demonstrate that GO accelerates the hydration process, due to its effect as a nucleation site. As indicated before, this impact increases the hydration product quantity, thus intensifying the cement microstructure and causing a decrease and change in the volume and structure of pores. It is also feasible that the morphology of the hydration products precipitated on GO can be altered, that is, they become smaller, allowing them to fill the holes. Therefore, the morphology of the hydration product should be studied later.

### 3.6. Chloride Diffusion in Mortars

The coefficient of chloride diffusion is one of the most critical durability factors, since it refers to chloride penetration into concrete, and hence, steel reinforcement corrosion. Figure 11 depicts the content of chloride ion vs. depth for cement mortar samples with varying amounts of GO. As observed from the curve, the chloride ion content of all cement mortar samples declines with increasing depth. Moreover, the total chloride content was greater for MG0 for all depths, except for the initial depth (MG1 was the largest). Similar results were found for the second type of mortar containing the SP. Compared with reference sample MGS0, all GO-modified cement mortar samples showed a lower amount of chloride content at all depths. It should also be noted that sample MGS1 showed a lower chloride content than sample MGS2 at almost all depths, while the MGS3 sample still had the best results in the sense of low chloride content at all depths.

In addition, in terms of comparing the two kinds of mortar, manufactured with or without SP, all samples manufactured with SP showed better results, meaning less chloride content, than those manufactured without SP. This observation can be explained by the efficient function of SP in enhancing GO dispersion in the cement matrix. Consequently, the findings of this research demonstrated that the incorporation of GO into cement mortar significantly increases its resistance to chloride ion penetration. This is consistent with past studies [26,49]. For the purpose of providing an explanation or a vision of the results obtained, it should be recalled that in the results of gas permeability, related to porosity, in the samples of modified cement mortar with GO, porosity decreased.

It is well known that porosity is the main factor determining the service life of cementitious materials. For example, the presence of higher porosity increases the vulnerability of concrete to chemical attacks, such as those from CO_2_ or Cl^−^. As revealed in the TGA and ^29^Si MAS-NMR results, GO can accelerate the hydration process because it has the advantage of providing an adsorption site where more hydration products are formed. As a result, an increase in the amount of hydration products formed changes the microstructure of the cement mortar and forms a dense cross-linked structure, with a reduction in pore volume, further preventing chloride ions from entering the cement mortar. It is also possible to make a correlation with the results of the electrical resistivity test, because the higher electrical resistivity indicates a higher resistance to aggressive attack.

## 4. Conclusions

According to the results of thermogravimetric analysis (TGA), at 7 days GO had no significant effects on the contents of both C–S–H and CH formed, while at 28 days sample CG3 (containing a moderate amount of GO) showed the largest content of C–S–H. The increase in the content of C–S–H compared to the reference sample was estimated at 5.46%, this is attributed to the fact that the GO acts as a nucleation site.The ^29^Si MAS-NMR findings were comparable with the TGA results. The ^29^Si MAS-NMR tests revealed that the addition of GO increased the hydration degree at advanced age, along with enhancing the main chain length value. The lengthening of the Main chain length (MCL) after the addition of GO could be due to the generation of more tetrahedron Q^2b^ on the GO surface area. Tetrahedron Q^2b^ is known to be bridge-shaped and can act as a point connection between short chains, generating a larger chain length.In mechanical strength results, GO was found to be more effective at advanced ages. With the addition of G3, at 28 days, an increase in the flexural resistance of both types of mortar was observed; the flexural strength of MG3 increased by 19.72% and MGS3 by 30.85%. It is worth noting the effective role of SP in achieving a good dispersion of GO. In terms of compressive strength, GO-containing samples achieved slight improvements; MG3 increased by 9.33% and MGS1 by 8.45% compared with the reference samples. This improvement was due to a variety of GO-specific enhancement mechanisms, such as an accelerated cement hydration process due to GO’s impact as a nucleation site.The results of durability tests conducted on 28-day cured GO-modified samples showed that they had significantly reduced the chloride ion content at all depths and oxygen penetration compared to the reference sample. This was attributed to GO catalyzing the hydration process, as evidenced by TGA and ^29^Si-NMR spectroscopy. The denser microstructure of the GO-modified samples, with a lower percentage of cracks and pores, resulted in greater resistance to penetration by aggressive agents.The enhanced and denser calcium silicate hydrate (C–S–H) gel structures, which block ionic conduction pathways, have also been observed in graphene oxide (GO)-modified cement mortar through the enhanced electrical resistivity values compared to reference samples.

## Figures and Tables

**Figure 1 materials-17-01445-f001:**
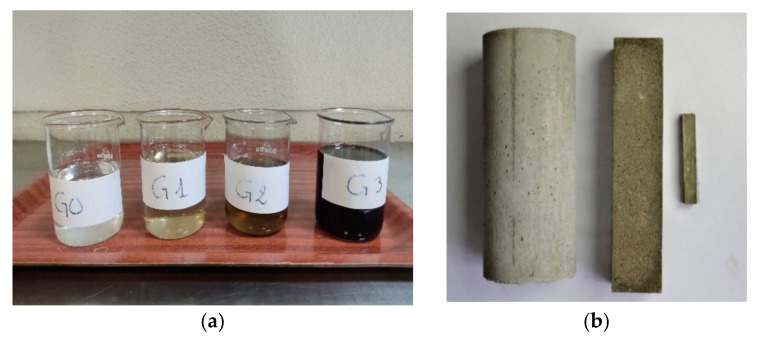
(**a**) The prepared GO–water suspensions. (**b**) Specimen types produced.

**Figure 2 materials-17-01445-f002:**
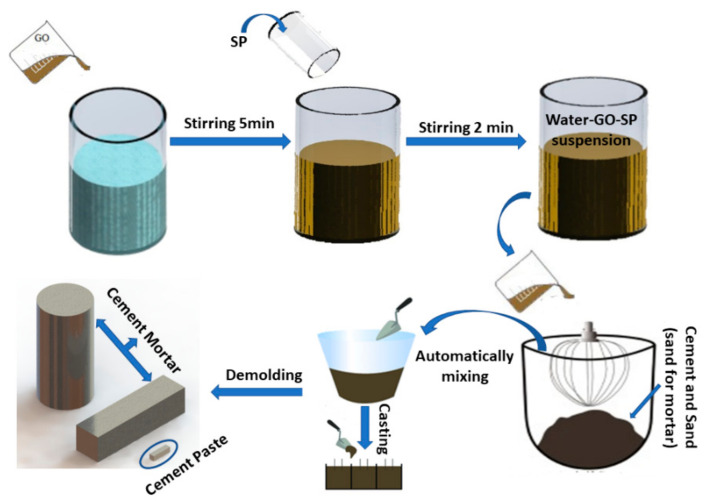
Schematic illustration of the preparation process for GO-modified cement composites.

**Figure 3 materials-17-01445-f003:**
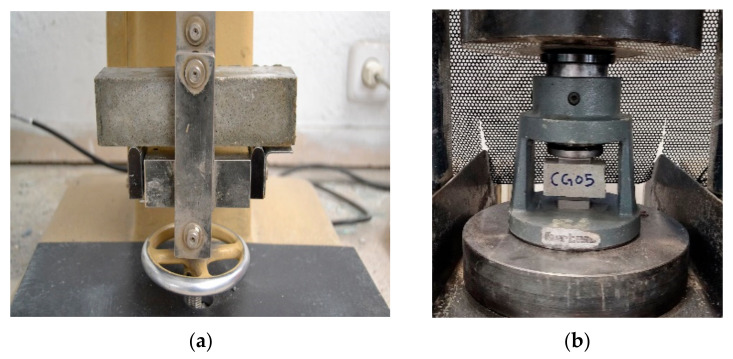
Tests of (**a**) flexural and (**b**) compressive strength.

**Figure 4 materials-17-01445-f004:**
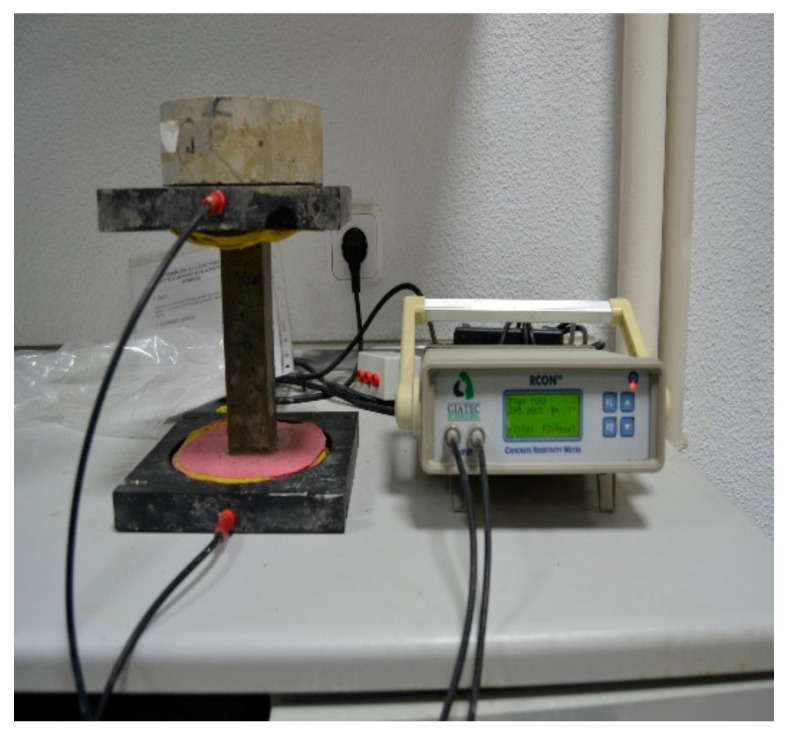
Electrical resistivity test.

**Figure 5 materials-17-01445-f005:**
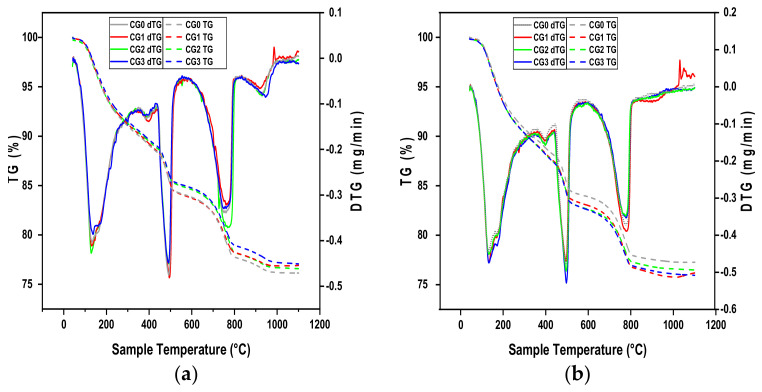
TGA and DTG curves of cement paste at (**a**) 7 days and (**b**) 28 days.

**Figure 6 materials-17-01445-f006:**
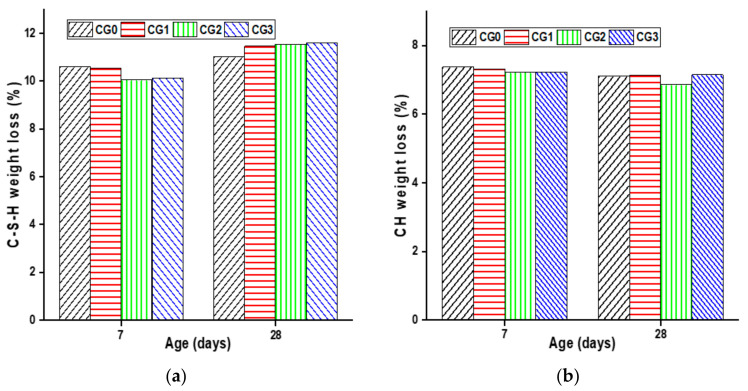
The hydration product content at 7 and 28 days: (**a**) C–S–H gel content and (**b**) CH Content.

**Figure 7 materials-17-01445-f007:**
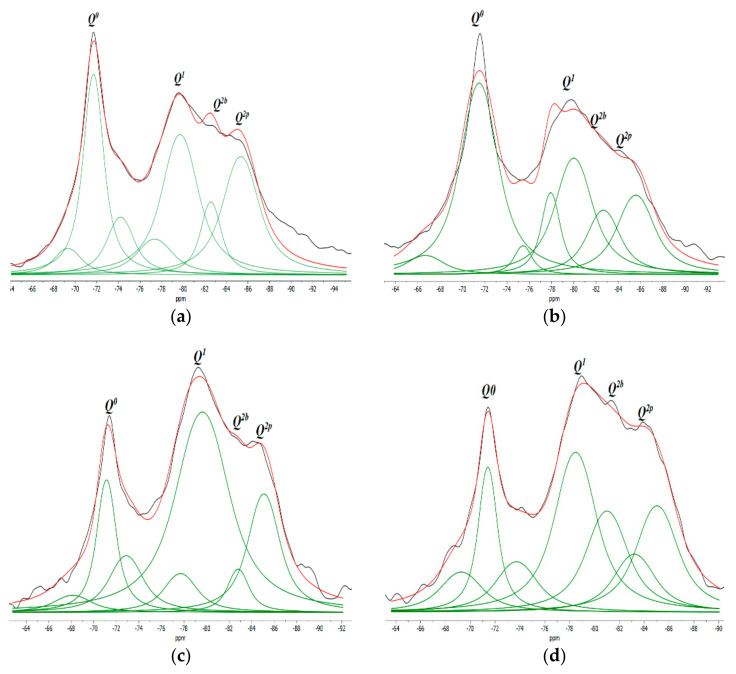
^29^Si MAS-NMR spectrums of (**a**) CG0 at 7 days; (**b**) CG3 at 7 days; (**c**) CG0 at 28 days; and (**d**) CG3 at 28 days.

**Figure 8 materials-17-01445-f008:**
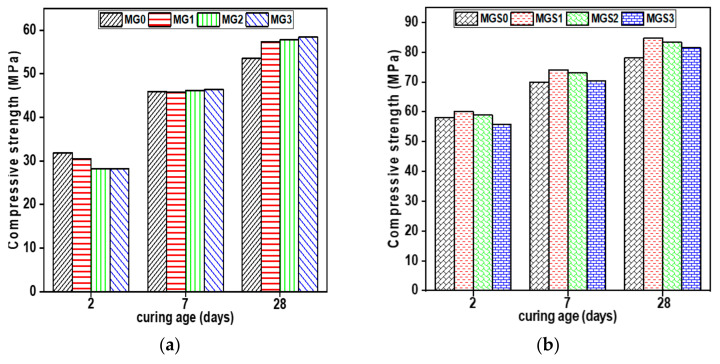
Compressive strength average of mortar samples (**a**) without SP and (**b**) with SP, and flexural strength average of mortar samples (**c**) without SP and (**d**) with SP.

**Figure 9 materials-17-01445-f009:**
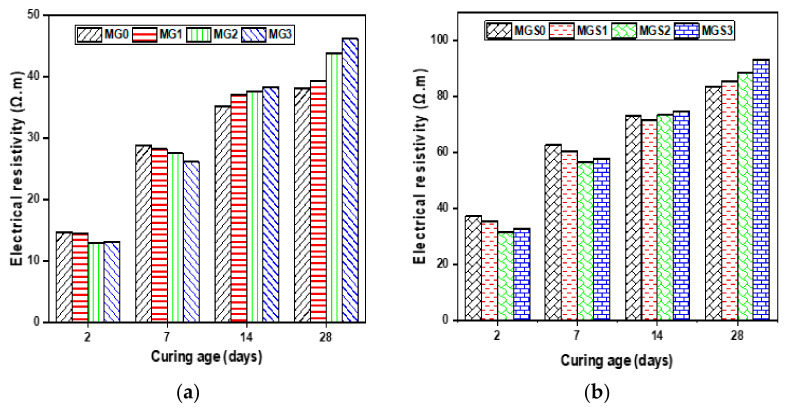
Electrical resistivity average of mortar samples (**a**) without SP and (**b**) with SP.

**Figure 10 materials-17-01445-f010:**
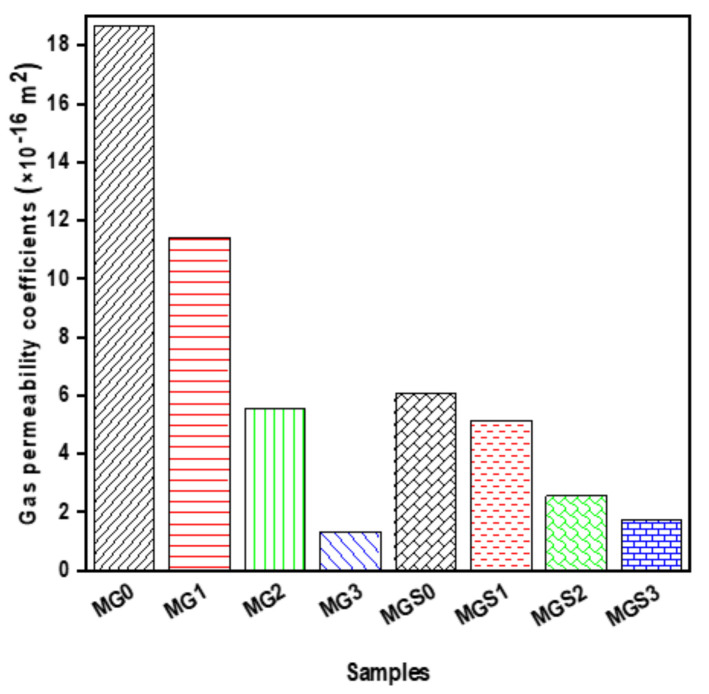
Gas permeability of mortars.

**Figure 11 materials-17-01445-f011:**
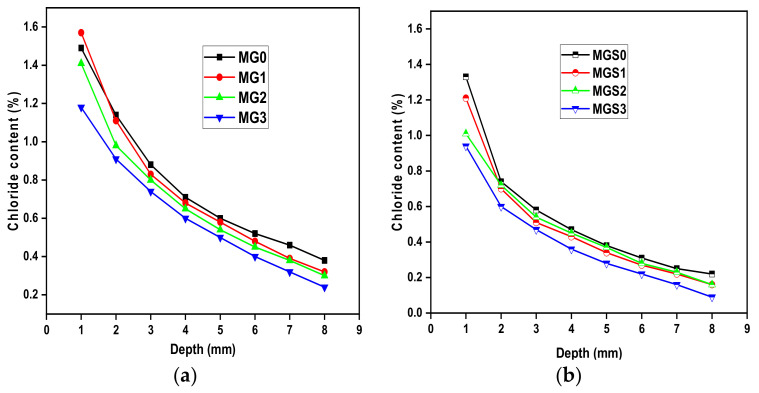
Total chloride ion content of cement mortars: (**a**) samples without SP and (**b**) samples with SP.

**Table 1 materials-17-01445-t001:** Comparison of various doses of GO on the strength and durability of cement composites in previous studies.

Matrix	*w*/*c* Ratio	GO %	GO Dispersion	Strength Results	Durability	References
Paste	0.50	0.05	Not specified	Compressive 24%/7d Flexural 41%/7d	Not specified	[17]
Concrete	0.45	0.02–0.08	Superplasticizer	Compressive 49%/90d Tensile 38%/90d	Reduced permeability	[30]
Mortar	0.37–0.60	0.10	Ultrasonication/superplasticizer	Compressive 15–33%	Resistance to sulfuric acid attack	[31]
Mortar	Not specified	0.02–0.10	Not specified	Compressive 10%	Reduced permeability	[32]
Mortar	0.34	0.01–0.06	/	/	Enhanced water sorptivity and chloride penetration	[26]

**Table 2 materials-17-01445-t002:** Chemical composition of Portland cement I 52.5 R.

CEM I	Chemical Composition (wt.%)							
	CaO	SiO_2_	Al_2_O_3_	SO_3_	Fe_2_O_3_	MgO	K_2_O	LOI ^1^
%	61.5	20.5	5.03	3.35	3.20	1.45	1.05	2.39

^1^ Loss on ignition.

**Table 3 materials-17-01445-t003:** Mix proportions of cement mortar.

Samples	Cement (g)	*w*/*c* *	GO	SP (wt.%)	Sand (g)
MG0	450	0.50	G0	-	1350
MG1	450	0.50	G1	-	1350
MG2	450	0.50	G2	-	1350
MG3	450	0.50	G3	-	1350
MGS0	450	0.35	G0	2.0	1350
MGS1	450	0.35	G1	2.0	1350
MGS2	450	0.35	G2	2.0	1350
MGS3	450	0.35	G3	2.3	1350

* The amount of total water used in our experiment was determined based on the water-to-cement ratio (*w*/*c*), with adjustments made to account for the presence of water in the GO and SP.

**Table 4 materials-17-01445-t004:** Mix proportions of cement paste.

Samples	Cement (g)	*w*/*c* *	GO
CG0	450	0.50	G0
CG1	450	0.50	G1
CG2	450	0.50	G2
CG3	450	0.50	G3

* The total water content was determined in accordance with the established water-to-cement (*w*/*c*) ratio. Subsequently, the water content associated with GO and SP was subtracted from the overall water content value.

**Table 5 materials-17-01445-t005:** Percentages of silicate chemical shift and parameters from deconvolution of ^29^Si MAS-NMR spectra.

Sample	Q^0^(%)	Q^1^ (%)	Q^2b^ (%)	Q^2p^ (%)	α (%)	MCL
7 days						
CG0	43.53	34.05	8.97	13.45	56.47	3.31
CG3	44.94	34.65	9.4	11.01	55.06	3.18
28 days						
CG0	28.56	46.81	9.59	15.04	71.44	3.05
CG3	26.70	44.76	12.06	16.48	73.30	3.28

## Data Availability

The data presented in this study are available on request from the corresponding author due to confidentiality reasons imposed by the company Graphenea.

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
