# Peer review of "Influence of Graphene Oxide on Mechanical Properties and Durability of Cement Mortar"

_materials, 2024, doi:10.3390/ma17061445_

Round 1

Reviewer 1 Report (Previous Reviewer 1)

Comments and Suggestions for Authors

The comments are not addressed and the paper still lacks novelty. Therefore, the paper should be rejected. 

Comments on the Quality of English Language

The paper is filled with grammatical errors. 

Author Response

We wish to extend our sincere gratitude to the reviewers for their insightful comments, which have assisted in rectifying certain issues and enhancing the final version of the manuscript.

In light of the observations provided by the reviewers, we present herein a comprehensive summary of the modifications made to the manuscript (highlighted in blue font in the revised version).

Yours sincerely,

  1. In Introduction, authors should add a Table that compares the GO, GO dose weight, preparation methods, mechanical and durability properties with published literatures

A part of the introduction conducts a comparative analysis of prior research endeavors, precisely the outcomes attained following the incorporation of varying graphene oxide concentrations. Furthermore, a table has been included, which enumerates recent studies and juxtaposes the effects of diverse graphene oxide dosages on durability and strength characteristics.

Table 1. Comparison of various doses of GO on the strength and durability of cement composites in previous studies

Matrix

w/c ratio

GO %

GO dispersion

Strength results

Durability

references

Paste

0.5

0.05

Not specified

Compressive 24%/7d

Flexural 41%/7d

Not specified

[17]

Concrete

0.45

0.02-0.08

superplasticizer

Compressive 49%/90d

Tensile 38%/90d

reduced permeability

[27]

Mortar

0.37-0.6

0.1

Ultrasonication/superplasticizer

Compressive 15-33%

resistance to sulfuric acid attack/

[28]

Mortar

Not specified

0.02-0.1

Not specified

Compressive 10%

reduced permeability

[29]

Mortar

0.34

0.01-0.06

/

/

Enhanced water sorptivity and chloride penetration

[23]

2. In Introduction, authors should explain a bit more about the novelty and importance of this study?

The study addresses the current lack of consensus on optimal graphene oxide (GO) dosage by employing a broader range of concentrations compared with previous research (in this study the GO dosage varies between the order 10-4 and 10-2). It investigates the impact of GO on cement hydration and microstructure across two water-to-cement ratios, shedding light on potential agglomeration effects. Furthermore, it employs nuclear magnetic resonance (NMR), a technique relatively underutilized in GO-cement studies, to further investigate the degree of hydration and the structure of gel chains, crucial factors influencing the ultimate strength of the composite material.

3. In Materials and methods, authors should add sample preparation schematic figure for better understanding

 As per your request, a schematic illustrating the mixing method for cement and graphene oxide has been incorporated and a copy has been attached here

Figure 2. Schematic illustration of the preparation process for GO-modified cement composites

4. In Thermal Gravimetric Analysis (TGA), what carrier gas used during TGA analysis?

Nitrogen was identified as the carrier gas used in the Thermal Gravimetric Analysis (TGA) analysis. This information has been incorporated into the manuscript on page 6, line 190-191, for clarity.

5. In Table 4, authors should explain why the Q0(%) is reduced and MCL is increased after 28 days for CG3?

It has been incorporated the following text into the manuscript. This new content can be found on page 10-11, lines 347-353.

It can also be seen from the results of table 4, that the diminution in the unhydrated Q0(%) and the augmentation of the MCL can be attributed to the inherent properties of Q0(%). Upon the liberation of free water absorbed by the GO, a hydration process accelerates, transforming the Q0(%) into a silicate polyhedron. This transformation facilitates the adhesion to pre-existing chains, a process made possible by the bridging capabilities of the GO. Consequently, this leads to an increase in the MCL. It can also be seen from the results of table 4, that the diminution in the unhydrated Q0(%) and the augmentation of the MCL can be attributed to the inherent properties of Q0(%). Upon the liberation of free water absorbed by the GO, a hydration process accelerates, transforming the Q0(%) into a silicate polyhedron. This transformation facilitates the adhesion to pre-existing chains, a process made possible by the bridging capabilities of the GO. Consequently, this leads to an increase in the MCL.)

6. In Introduction section, authors should add more GO properties details or paragraph on GO. Authors may go through these publications for more details and cite accordingly:https://doi.org/10.1016/j.cartre.2023.100251 , https://doi.org/10.1016/j.apsusc.2019.05.321 & https://doi.org/10.1016/j.carbon.2023.118447

Thank you for your suggestion. References mentioned here have been included in the Introduction section

Reviewer 2 Report (Previous Reviewer 3)

Comments and Suggestions for Authors

Thank you for your appropriate response to the peer review comments. I recommend that it be accepted for publication.

Author Response

Dear reviewer,

Thank you for your kind comments.

Best regards,

The authors

Reviewer 3 Report (New Reviewer)

Comments and Suggestions for Authors

The authors in the present manuscript show that the effect of graphene oxide (GO) on the mechanical strengths and durability of cement composites was investigated by preparing GO-modified cement mortars. Thermogravimetric analysis (TGA) and nuclear magnetic resonance (29Si MAS-NMR) were performed on the cement paste to investigate the influence of GO on the hydration process and chain structure of calcium-silicate-hydrate (C-S-H) gels. Additionally, mechanical strength tests, such as compressive and flexural, were conducted on cement mortars. The optimal dosage of GO increased the compressive strength by 9.02% after 28 days. Furthermore, the flexural strength of cement mortars with the combination of GO and superplasticizer (SP) after days increased by 21.86%, compared with reference mortar. The impact of GO proved to be more pronounced and beneficial in the durability tests, suggesting that GO can enhance the microstructure through hydration products to create a dense and interconnected microstructure. The authors should address the following issues and information’s before publication acceptance in the prestigious ‘Materials’ Journal:

1. In Introduction, authors should add a Table that compares the GO, GO dose weight, preparation methods, mechanical and durability properties with published literatures.

2. In Introduction, authors should explain a bit more about the novelty and importance of this study?

3. In Materials and methods, authors should add sample preparation schematic figure for better understanding?

4. In Thermal Gravimetric Analysis (TGA), what carrier gas used during TGA analysis?

5. In Table 4, authors should explain why the Q0(%) is reduced and MCL is increased after 28 days for CG3?

6. In Introduction section, authors should add more GO properties details or paragraph on GO. Authors may go through these publications for more details and cite accordingly: https://doi.org/10.1016/j.cartre.2023.100251 , https://doi.org/10.1016/j.apsusc.2019.05.321 & https://doi.org/10.1016/j.carbon.2023.118447  

Comments on the Quality of English Language

Minor editing of English language required.

Author Response

Dear reviewer,

We wish to extend our sincere gratitude to the reviewers for their insightful comments, which have assisted in rectifying certain issues and enhancing the final version of the manuscript.

In light of the observations provided by the reviewers, we present herein a comprehensive summary of the modifications made to the manuscript (highlighted in blue font in the revised version).

Yours sincerely,

The authors

  1. In Introduction, authors should add a Table that compares the GO, GO dose weight, preparation methods, mechanical and durability properties with published literatures

    A part of the introduction conducts a comparative analysis of prior research endeavors, precisely the outcomes attained following the incorporation of varying graphene oxide concentrations. Furthermore, a table has been included, which enumerates recent studies and juxtaposes the effects of diverse graphene oxide dosages on durability and strength characteristics.

    Table 1. Comparison of various doses of GO on the strength and durability of cement composites in previous studies

    Matrix

    w/c ratio

    GO %

    GO dispersion

    Strength results

    Durability

    references

    Paste

    0.5

    0.05

    Not specified

    Compressive 24%/7d

    Flexural 41%/7d

    Not specified

    [17]

    Concrete

    0.45

    0.02-0.08

    superplasticizer

    Compressive 49%/90d

    Tensile 38%/90d

    reduced permeability

    [27]

    Mortar

    0.37-0.6

    0.1

    Ultrasonication/superplasticizer

    Compressive 15-33%

    resistance to sulfuric acid attack/

    [28]

    Mortar

    Not specified

    0.02-0.1

    Not specified

    Compressive 10%

    reduced permeability

    [29]

    Mortar

    0.34

    0.01-0.06

    /

    /

    Enhanced water sorptivity and chloride penetration

    [23]

    2. In Introduction, authors should explain a bit more about the novelty and importance of this study?

The study addresses the current lack of consensus on optimal graphene oxide (GO) dosage by employing a broader range of concentrations compared with previous research (in this study the GO dosage varies between the order 10-4 and 10-2). It investigates the impact of GO on cement hydration and microstructure across two water-to-cement ratios, shedding light on potential agglomeration effects. Furthermore, it employs nuclear magnetic resonance (NMR), a technique relatively underutilized in GO-cement studies, to further investigate the degree of hydration and the structure of gel chains, crucial factors influencing the ultimate strength of the composite material.

3. In Materials and methods, authors should add sample preparation schematic figure for better understanding

As per your request, a schematic illustrating the mixing method for cement and graphene oxide has been incorporated and a copy has been attached here

Figure 2. Schematic illustration of the preparation process for GO-modified cement composites

4. In Thermal Gravimetric Analysis (TGA), what carrier gas used during TGA analysis?

Nitrogen was identified as the carrier gas used in the Thermal Gravimetric Analysis (TGA) analysis. This information has been incorporated into the manuscript on page 6, line 190-191, for clarity.

5. In Table 4, authors should explain why the Q0(%) is reduced and MCL is increased after 28 days for CG3?

It has been incorporated the following text into the manuscript. This new content can be found on page 10-11, lines 347-353.

It can also be seen from the results of table 4, that the diminution in the unhydrated Q0(%) and the augmentation of the MCL can be attributed to the inherent properties of Q0(%). Upon the liberation of free water absorbed by the GO, a hydration process accelerates, transforming the Q0(%) into a silicate polyhedron. This transformation facilitates the adhesion to pre-existing chains, a process made possible by the bridging capabilities of the GO. Consequently, this leads to an increase in the MCL. It can also be seen from the results of table 4, that the diminution in the unhydrated Q0(%) and the augmentation of the MCL can be attributed to the inherent properties of Q0(%). Upon the liberation of free water absorbed by the GO, a hydration process accelerates, transforming the Q0(%) into a silicate polyhedron. This transformation facilitates the adhesion to pre-existing chains, a process made possible by the bridging capabilities of the GO. Consequently, this leads to an increase in the MCL.)

6. In Introduction section, authors should add more GO properties details or paragraph on GO. Authors may go through these publications for more details and cite accordingly:https://doi.org/10.1016/j.cartre.2023.100251 , https://doi.org/10.1016/j.apsusc.2019.05.321 & https://doi.org/10.1016/j.carbon.2023.118447

Thank you for your suggestion. References mentioned here have been included in the Introduction section

Reviewer 4 Report (New Reviewer)

Comments and Suggestions for Authors

Overall, the study presents intriguing insights into the impact of graphene oxide on cement mortar, although there are several points that require clarification and improvement before final presentation.

1) In the abstract, please provide more detailed commentary on the results obtained in the study. This will offer readers a clearer understanding of the significance of your results.

2) The 29Si MAS-NMR deconvolutions should be redone and more detail on this point should be provided in the text. All data presented in Table 4 must be modified after improved of the deconvoluted spectra.

3) Please include contact angle analysis (e.g., Catalysts 2023, 13(12), 1479 / International Journal of Pavement Engineering, 2020, 21(14), 1746) for all samples prepared in this study. Briefly comment on these results and compare them with the literature. Also, I recomend that the suggested references must be included in the study.

4) An analysis of variance (ANOVA) of the data obtained in this study would be beneficial for evaluating the significance of any observed differences. Including ANOVA will enhance the statistical robustness of your findings and provide valuable insights into the effects of graphene oxide addition on the mechanical properties and durability of cement mortar.

Addressing these points will significantly improve the quality and comprehensiveness of your manuscript.

Author Response

Dear reviewer,

We wish to extend our sincere gratitude to the reviewers for their insightful comments, which have assisted in rectifying certain issues and enhancing the final version of the manuscript.

In light of the observations provided by the reviewers, we present herein a comprehensive summary of the modifications made to the manuscript (highlighted in blue font in the revised version).

Yours sincerely,

The authors

1. In the abstract, please provide more detailed commentary on the results obtained in the study. This will offer readers a clearer understanding of the significance of your results  

The authors would like to clarify that the abstract has been updated and includes all results tests conducted in this study and commentary on their significance. For your convenience, an updated abstract is attached here for your perusal, and the added parts are highlighted in blue for ease of identification.

Abstract: The effect of graphene oxide (GO) on the mechanical strengths and durability of cement composites was reserached by preparing GO-modified cement mortars. Thermogravimetric analysis (TGA) and nuclear magnetic resonance (29Si MAS-NMR) were performed on the cement paste to investigate the influence of GO on the hydration process and chain structure of calcium-silicate-hydrate (C-S-H) gels. TGA revealed that the high GO dosage increased the content of C-S-H by 5.46% compared with the control at 28 days. Similarly, 29Si-NMR improved the hydration degree and main chain length (MCL) in GO-modified samples at 28 days. The GO led to increases of 2.54% and 7.01% in the hydration degree and MCL, respectively, compared with the control at 28 days. These findings underscore the multifaceted role of GO in improving the mechanical properties and durability of cement composites. Mechanical strength tests, such as compressive and flexural, were conducted on cement mortars. The optimal dosage of GO increased the compressive strength by 9.02% after 28 days. Furthermore, the flexural strength of cement mortars with the combination of GO and superplasticizer (SP) after 28 days increased by 21.86%, compared with reference mortar. The impact of GO proved to be more pronounced and beneficial in the durability tests, suggesting that GO can enhance the microstructure through hydration products to create a dense and interconnected microstructure.

2. The 29Si MAS-NMR deconvolutions should be redone and more detail on this point should be provided in the text. All data presented in Table 4 must be modified after improved of the deconvoluted spectra.

The authors appreciate the reviewer's suggestion regarding the 29Si MAS-NMR deconvolutions. The spectra for one representative sample were carefully re-evaluated, with additional deconvolutions performed to ensure the accuracy of the data presented in Table 4. This re-evaluation, including a comparison to the original data (attached), confirmed the initial findings and did not reveal significant discrepancies

(a)

(b)

(a)  The old spectre; (b) the new spectre of sample CG0 at 7 days

3. Please include contact angle analysis (e.g., Catalysts 2023, 13(12), 1479 / International Journal of Pavement Engineering, 2020, 21(14), 1746) for all samples prepared in this study. Briefly comment on these results and compare them with the literature. Also, I recommend that the suggested references must be included in the study

The study acknowledges the importance of contact angle analysis as a valuable technique for investigating permeability, as suggested by the reviewer. Due to limitations in equipment availability, the technique was not incorporated into the current study. However, the study acknowledges its potential for future investigations to gain a more comprehensive understanding of the material's properties.

4. An analysis of variance (ANOVA) of the data obtained in this study would be beneficial for evaluating the significance of any observed differences. Including ANOVA will enhance the statistical robustness of your findings and provide valuable insights into the effects of graphene oxide addition on the mechanical properties and durability of cement mortar.

Regarding your suggestion to include an analysis of variance (ANOVA) to evaluate the significance of observed differences in our study, statistical analysis, including ANOVA, could indeed offer valuable insights. Our research design and objectives are primarily centered around experimental outcomes. Therefore, our study does not delve deeply into statistical analysis but rather focuses on experimental observations and their implications for practical applications.

Round 2

Reviewer 1 Report (Previous Reviewer 1)

Comments and Suggestions for Authors

The paper can be accepted. 

This manuscript is a resubmission of an earlier submission. The following is a list of the peer review reports and author responses from that submission.

Round 1

Reviewer 1 Report

Comments and Suggestions for Authors

This study investigated the impact of graphene oxide (GO) on cement composite strength and durability. GO-modified cement mortars were tested, revealing increased compressive and flexural strength with 0.05 wt.% GO addition. However, before further consideration of the manuscript, the authors must “fully” address the comments listed below:

1.      Could you elaborate on the thermogravimetric analysis (TGA) and nuclear magnetic resonance (29Si MAS-NMR) techniques used to investigate the influence of GO on the hydration process and chain structure of calcium-silicate-hydrate (C-S-H) gels in cement paste?

2.      Can you provide more details on the flexural strength improvements observed in cement mortars with 0.05 wt.% GO and superplasticizer (SP) after 7 and 28 days, and how these improvements compared with the reference mortar?

3.      How was the dispersion of GO in cement matrix addressed to prevent agglomeration, and why was it dissolved in mixing water first? Could you elaborate on the mixing process and the inclusion of superplasticizer (SP)?

4.      What were the specifics of the 29Si Magic Angle Spinning Nuclear Magnetic Resonance (29Si MAS-NMR) analysis performed on crushed cement paste samples at 7 and 28 days, and how were the chemical shifts of 29Si referenced and processed?

5.      Could you elaborate on the method used for measuring the electrical resistivity of prismatic mortar specimens and the equations involved in calculating the electrical resistivity?

6.      You may mention in the introduction of the paper that apart from experimental analysis, the strength of mortars reinforced with graphene (e.g., figure 7 of your manuscript) can also be estimated via many micromechanics models such as Halpin-Tsai (https://doi.org/10.1007/s10999-022-09632-7) and Checkerboard (https://doi.org/10.1016/j.compstruct.2019.01.041) models. You can reference the referred papers in your manuscript. 

7.      Based on the research findings, what conclusions can be drawn regarding the effect of GO on the mechanical properties and durability of mortar, specifically in terms of thermogravimetric analysis results, 29Si MAS-NMR findings, and the impact on mechanical strength and durability tests?

Reviewer 2 Report

Comments and Suggestions for Authors

Paper ID: materials-2559618

Type:Article
Title: Effect of graphene oxide on mechanical properties and durability of cement mortar

This paper investigates the effect of graphene oxide on the mechanical properties and durability of cement mortar. Although the testing methods and compared results attained in the present study show the importance of the paper,

1.      No contribution to the literature.

2.      Topic is not original.

3.      Paper does not add any new information.

4.      Novelty in comparison to recent literature is missing.

5.      There are many studies on this subject. you can find some recent studies published in MDPI journals:

- Ganesh, S., Thambiliyagodage, C., Perera, S. J., & Rajapakse, R. K. N. D. (2022). Influence of Laboratory Synthesized Graphene Oxide on the Morphology and Properties of Cement Mortar. Nanomaterials13(1), 18.

-Wang, Y., Yang, J., & Ouyang, D. (2019). Effect of graphene oxide on mechanical properties of cement mortar and its strengthening mechanism. Materials12(22), 3753.

- Sun, H., Ling, L., Ren, Z., Memon, S. A., & Xing, F. (2020). Effect of graphene oxide/graphene hybrid on mechanical properties of cement mortar and mechanism investigation. Nanomaterials10(1), 113. 

Reviewer 3 Report

Comments and Suggestions for Authors

This paper discusses on the effect graphene oxide on mechanical properties and durability of cement mortar. The topic is interesting and warrants a constructive discussion on the improvement method of concrete materials using nanomaterials. However, several points as indicated below need to be addressed by authors to improve the quality of the articles.

“1.Introduction” and “4. Conclusions”

The novelty and originality of this study is not clear; please clarify this in chapter 1. I also think that the summary should be written with a comparison with previous reports, emphasizing the newly obtained findings.

p.3, Line114-116, Fig.1(a)

Please specify the amount of GO added for each solution in the figure. It cannot be determined from the photo.

p.3, Line119-121

Please specify the rationale for selecting the dosage of GO.

p.5, Line157-158

Please specify the particle size of the powder and the storage method (temperature, humidity, etc.) after pulverization.

p.11, Line345-347

The trend of compressive strength differs depending on whether SP is present or not; with SP, the strength of MGS0005 is greater than that of MGS05. Please provide your own perspective on the difference in compressive strength trends due to the addition of SP.

p.11, Fig.7

The caption((a)-(d)) is positioned away from the figure, please move it closer to the relevant figure.

p.9, Fig.9

The left-most sample name is incorrect. Replace "MG05" with "MG0".